# The InnoRec Process: A Comparative Study of Three Mainstream Routes for Spent Lithium-ion Battery Recycling Based on the Same Feedstock

**Hao Qiu** [1,*], **Daniel Goldmann** [1], **Christin Stallmeister** [2], **Bernd Friedrich** [2], **Maximilian Tobaben** [3], **Arno Kwade** [3], **Christoph Peschel** [4], **Martin Winter** [4,5], **Sascha Nowak** [4], **Tony Lyon** [6] and **Urs A. Peuker** [6]

1. Institute of Mineral and Waste Processing, Recycling and Circular Economy Systems (IFAD), Clausthal University of Technology, Walther-Nernst-Straße 9, 38678 Clausthal-Zellerfeld, Germany; daniel.goldmann@tu-clausthal.de
2. Institute of Process Metallurgy and Metal Recycling (IME), RWTH Aachen University, Intzestr. 3, 52056 Aachen, Germany; cstallmeister@metallurgie.rwth-aachen.de (C.S.); bfriedrich@metallurgie.rwth-aachen.de (B.F.)
3. Institute for Particle Technology (iPAT), TU Braunschweig, Volkmaroder Str. 5, 38104 Braunschweig, Germany; m.tobaben@tu-braunschweig.de (M.T.); a.kwade@tu-braunschweig.de (A.K.)
4. MEET Battery Research Center, University of Münster, Corrensstr. 46, 48149 Münster, Germany; christoph.peschel@uni-muenster.de (C.P.); martin.winter@uni-muenster.de (M.W.); sascha.nowak@uni-muenster.de (S.N.)
5. Helmholtz-Institute Münster, IEK-12, Forschungszentrum Jülich, Corrensstraße 46, 48149 Münster, Germany
6. Institute of Mechanical Process Engineering and Mineral Processing (MVTAT), TU Bergakademie Freiberg, 09599 Freiberg, Germany; tony.lyon@mvtat.tu-freiberg.de (T.L.); urs.peuker@mvtat.tu-freiberg.de (U.A.P.)
* Correspondence: hao.qiu@tu-clausthal.de

**Abstract:** Among the technologies used for spent lithium-ion battery recycling, the common approaches include mechanical treatment, pyrometallurgical processing and hydrometallurgical processing. These technologies do not stand alone in a complete recycling process but are combined. The constant changes in battery materials and battery design make it a challenge for the existing recycling processes, and the need to design efficient and robust recycling processes for current and future battery materials has become a critical issue today. Therefore, this paper simplifies the current treatment technologies into three recycling routes, namely, the hot pyrometallurgical route, warm mechanical route and cold mechanical route. By using the same feedstock, the three routes are compared based on the recovery rate of the six elements (Al, Cu, C, Li, Co and Ni). The three different recycling routes represent specific application scenarios, each with their own advantages and disadvantages. In the hot pyrometallurgical route, the recovery of Co is over 98%, and the recovery of Ni is over 99%. In the warm mechanical route, the recovery of Li can reach 63%, and the recovery of graphite is 75%. In the cold mechanical route, the recovery of Cu can reach 75%, and the recovery of Al is 87%. As the chemical compositions of battery materials and various doping elements continue to change today, these three recycling routes could be combined in some way to improve the overall recycling efficiency of batteries.

**Keywords:** battery recycling; lithium; graphite; electrolyte; mechanical processing; pyrometallurgy; thermal treatment; pyrolysis; flotation; hydrometallurgy

## 1. Introduction

With the gradual increase in the popularity of electric vehicles worldwide, the production of lithium-ion batteries (LIBs) is experiencing incredible growth. The effective management and recycling of end-of-life (EoL) LIBs is becoming a crucial issue for battery sustainability, and as a result, this area is gaining considerably more attention than before.

A LIB module can be mainly divided into a cathode, anode, electrolyte, separator, cell housing and module periphery. Considering the cost and electrochemical performance,

cathode active materials formed by mixed transition-metal oxides such as $LiNi_xCo_yMn_zO_2$ have become a vital direction and are widely used in many electric vehicle models to replace $LiMnO_2$ [1,2].

From the perspective of urban mineral resources, EoL-LIBs contain a large number of valuable components. For example, NMC cathode active materials and their foils are rich in Ni, Co, Mn, Li and Al, and anode materials and their foils are rich in graphite and Cu. The module periphery mainly contains Fe and Al [1]. As the prices of these raw materials have increased significantly over the past few years (Table 1), and as the European Commission has added Ni, Co, Cu, Al, Li and graphite to the list of critical and strategic raw materials [3], the recovery of valuable components from LIBs is of strategic importance in securing the supply chain and is an important part of the circular economy.

**Table 1.** The price of selected materials in lithium-ion battery raw materials.

| Product | Average Price 2019, US Dollar/mt | Average Price 2020, US Dollar/mt | Average Price 2021, US Dollar/mt | Average Price 2022, US Dollar/mt | References |
|---|---|---|---|---|---|
| Ni | 13,903 | 13,772 [4] | 18,000 | 25,000 | [4] |
| Co | 37,368 | 34,612 | 50,706 | 68,343 | [5] |
| Al | 1794 | 1704 | 2473 | 2705 | [6,7] |
| Cu | 6010 | 6174 | 9317 | 8822 | [6,7] |
| $Li_2CO_3$ (battery grade) | 11,700 | 8400 | 12,600 | 37,000 | [8] |
| Graphite (flake) | 1340 | 1340 | 1390 | 1300 | [9] |

Currently, the recycling technologies for EoL-LIBs include high-temperature pyrometallurgical treatment, mid/low-temperature thermal treatment such as pyrolysis and roasting, mechanical treatment and hydrometallurgical treatment [1,10–16]. In a complete recycling process, these pathways do not exist separately but are combined in a certain way. Each of these recycling routes has its own application scenarios. For instance, the advantages of pyrometallurgical recovery technology are its high processing capacity and the fact that it does not require the mechanical pretreatment of its feed. On the other hand, mechanical pre-treatment combined with hydrometallurgical processing is more capable of recovering a wider range of components [17].

Among several typical industrial recycling processes, the Umicore UHT process is a combined pyrometallurgical–hydrometallurgical process, the ACCUREC process is a combined pyrolysis–mechanical–hydrometallurgical process, and the Duesenfeld process is a combination of mechanical treatment and hydrometallurgical processing [18,19].

However, the application of new battery materials and new battery designs poses a great challenge to existing recycling processes [11]. In order to further optimize and integrate the existing recycling processes, the application-oriented research project InnoRec, funded by the German Federal Ministry of Education and Research (BMBF), aims to develop a holistic approach for the efficient and robust recycling of modern and future batteries. As a part of the project, this paper compared three representative recycling routes (Figure 1): the hot pyrometallurgical route (HP route, red), the warm mechanical route (WM route, orange), and the cold mechanical route (CM route, blue). In addition, the same feed (LIBs module, NMC-622) was used for all three recycling routes, and six elements (Al, Cu, C, Li, Co and Ni) were selected to compare their recoveries. Concerning the recycling of new battery materials, e.g., solid electrolytes, some explorations have also been conducted in the InnoRec project; refer to [10].

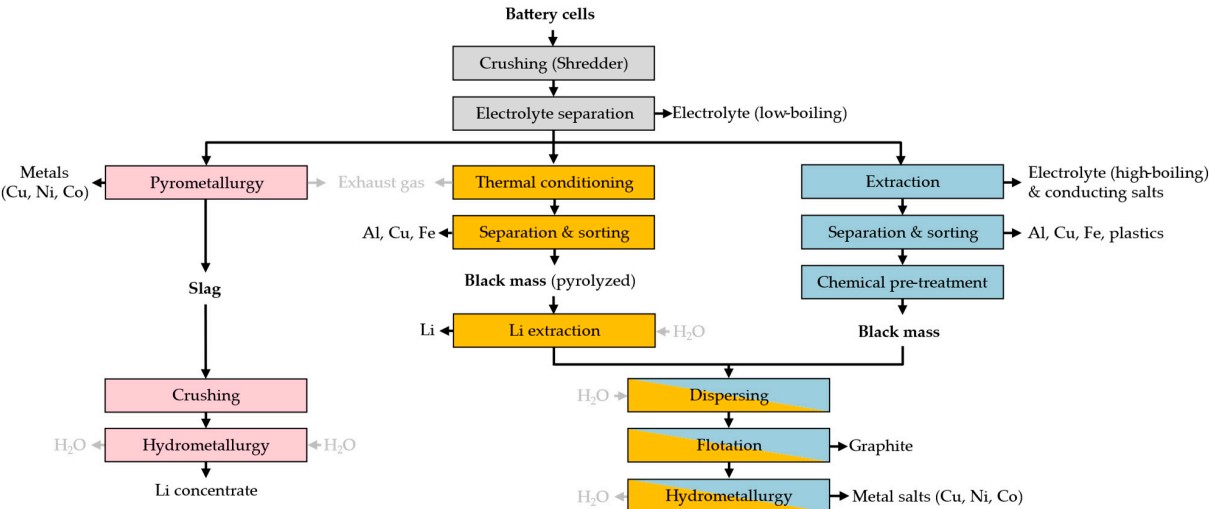

**Figure 1.** The three technological recycling routes of InnoRec.

For the extraction of electrolyte, although a self-made NMC-622 18,650 battery was used, this system could effectively ensure sample comparability to study a co-solvent's influence on carbonate and salt recovery.

## 2. Background: Recycling Technologies

### 2.1. High-Temperature Pyrometallurgical Processing

One recycling option for LIBs is a pyrometallurgical smelting process. Whole battery cells or the black mass (BM) are treated at high temperatures up to 1800 °C together with fluxes to recover Ni, Co and Cu in a metal alloy [17,20,21]. Oxygen-affine metals such as Al and Li are transferred in a slag phase. In the case of Li, evaporation and collection in the flue dust is also possible [17,20]. The organics and the graphite are used as reducing agents and energy sources for smelting [17]. Accordingly, the melting process is suitable as a splitting operation between different LIB elements [21]. This operation is very robust against fluctuating input streams, such as different battery chemistries, and shows high recovery rates for the Ni, Cu and Co noble metals of >95% [21]. Therefore, pyrometallurgical recycling is carried out industrially, for example, by Umicore and Nickel Hütte Aue [21,22].

However, so far, the focus in industry has been exclusively on the recovery of Ni, Co and Cu. A method with which to carry out the industrial recycling of Li from slag is not known. Since Li recovery has become crucial in LIB recycling today, a few studies have been carried out regarding its behavior during the smelting process, and possible recovery methods are being investigated. Sommerfeld et al. [23] compared Li enrichment in flue dust (80.4%) with Li enrichment in slag (82.4%) by smelting LIB BM in an electric arc furnace with different flux additions. Although similar enrichments were achieved in both cases, higher temperatures of 1800 °C vs. 1600 °C were necessary for volatilization. To recover Li either from slag or from flue dust, a hydrometallurgical leaching step is necessary. In the literature, just a few studies are known on Li recovery from slag. Klimko et al. leached a $SiO_2$-$Al_2O_3$-$Li_2O$-based slag with $H_2SO_4$ for Li recovery [24]. What was challenging was the high $SiO_2$ content in common slag systems due to silica gel formation, but this was overcome by a dry digestion process. In this case, nearly 100% Li leaching efficiency (LE) from the slag was reported.

### 2.2. Pyrolysis

Thermal pre-treatment steps are carried out for different kinds of organic-containing waste streams such as electronic scraps or batteries. This involves the treatment of the feed material at elevated temperatures under a defined atmosphere [25]. In the case of pyrolysis, the treatment is carried out under inert atmosphere ($N_2$ or Ar) or vacuum [25]. The aim is

to separate the organics contained in the feed by evaporation and cracking but to avoid uncontrolled exothermal reactions and melting of the metal content [21,26]. Therefore, the temperature for the treatment of LIBs is limited by the melting point of Al (660 °C). The process step can be carried out before or after shredding steps of the batteries and offers, in case of whole cells, the opportunity for a controlled and safe deactivation of the battery cells [17,21,26].

The removal of organics from the battery material is beneficial for following process steps such as flotation and hydrometallurgical treatment, as they can lower leaching kinetics and efficiencies and also influence the hydrophilic behavior of the metal oxides in the BM negatively [27–30]. Additionally, by evaporation and cracking reactions of the organics, a reducing atmosphere forms, which leads to the reduction in NMC metal oxides in the BM at elevated temperatures [26,31,32]. This offers the opportunity for water-based early-stage Li separation by phase transformation to water-soluble compounds, such as $Li_2CO_3$ [26,32,33]. Different studies demonstrate the high potential of a water-leaching step for selective Li recovery before entering further metallurgical recycling steps to avoid Li losses [26,32,33]. This requires sufficient process parameter adjustment in the thermal treatment process.

During the thermal treatment process of LIBs, an organic and fluorine-containing off-gas is produced [26,34]. Therefore, a sufficient gas-cleaning facility is necessary. There are only a few detailed studies on off-gas in the literature. But the understanding, control and influence of formed compounds is crucial for the process design.

### 2.3. Electrolyte Extraction

Accounting for double-digit wt% of state-of-the-art LIBs, the electrolyte is an often-overlooked component in the attempt to reach high recycling rates. Consisting of salt and solvents, electrolytes are not to be overseen for spent batteries. With a significant number of decomposition species, different physiochemical properties have to be addressed for targeted electrolyte recovery. While being burned and removed in direct thermal processes, volatile carbonates are recovered, e.g., by evaporation using elevated temperatures and/or reduced pressure during or after shredding. However, this removal is far from being quantitative, and further, regaining the electrolyte as such also includes salt recovery, ideally in a single processing step. Herein, the limited thermal stability of commercially applied $LiPF_6$ also has to be considered for usage in harsher thermal conditions.

A milder and potentially more efficient process for both the removal and targeted recovery of electrolytes is their extraction. The first literature reports on LIB recycling suggested liquid extractions in reasonably volatile solvents [35,36], while various further reports excluded electrolyte recovery [37,38]. Since liquid solvent extraction is expensive and regaining solvent fractions is complex and expensive, Sloop et al. patented the use of supercritical fluids and considered LIB electrolyte extraction by $CO_2$ [39]. Having an easily reachable supercritical phase (~31 °C and 74 bar), $CO_2$ can be used as a low-viscose solvent whose solvation properties can be further adjusted by the application of co-solvents. In contrast to liquid solvent extractions, the removal and recovery of the processing solvent can be easily obtained by pressure relaxation.

In academia, Liu et al. [40,41] and Grützke et al. [42,43] took a deeper look into extracting LIB electrolytes using super and subcritical $CO_2$ and elaborated parameters like pressure, temperature and time for static and continuous-flow extractions reaching > 85% electrolyte recovery. Further reports on the ($CO_2$) extraction of LIBs rather focused on metal or binder extraction, e.g., by the usage of complexing agents, than on the optimization and understanding of electrolyte recovery.

### 2.4. Mechanical Separation and Sorting Process

In the primary and secondary raw materials sectors, mechanical processing is based on breaking composite materials at their submaterial interfaces and the sorting of the respective materials. Depending on the composites, different stress mechanisms, such as cutting, compressive or impact stress, have to be used for the liberation comminution. The

particles are then sorted according to physical properties such as magnetism, electrical conductivity or density.

In the case of Li-ion batteries, the risk of thermal runaway must be taken into account during crushing. Therefore, special process parameters, such as an inert shredding chamber and discharged batteries, must be observed.

Valuable metals such as Ni and Co are found in the coatings of electrode foils. Therefore, it is an essential step to de-coat the foils through targeted mechanical stress. This results in a fine powder (also called black mass), which can be separated from the rest after detachment by sieve classification.

Purely by mechanical processes, it is generally impossible to produce material qualities that allow direct reuse in high-performance batteries, for example. For this reason, a combination of mechanical processing and metallurgy is always used industrially.

The thermal pre-treatment of batteries has a great influence on the mechanical treatment. Although the risk of thermal runaway is eliminated, the adhesive forces of the electrode coatings on the electrode foils are also changed. In addition, incomplete pyrolysis or a wrong temperature range can lead to agglomerates caused by melted plastics or a higher impurity amount in the fine fraction. Therefore, the parameters of a thermal pre-treatment must not only be investigated for each cell type, but rather, the effort and the benefit must also be compared with a "cold" route [44].

### 2.5. Ball Milling and Fine Grinding

Fine-grinding processes with target particle sizes in the single-digit micrometer range are essential in various industries, such as in the area of mineral processing, to ensure high product qualities and yields. In the context of recycling LIBs, mechanical comminution processes in grinding media mills are increasingly coming into focus, especially in research, in order to counteract the major disadvantages of hydrometallurgical process steps, such as a high demand for thermal energy and corrosive leaching agents. The objective is to enhance the LE of particulate systems by mechanically stressing intermediate recycling products, such as BM or slags, leading to a reduction in particle size and alterations in crystal structure.

This approach was tested, for example, by Guan et al. [45], who used $LiCoO_2$ powders from spent LIB as a feed material in their study. By processing the $LiCoO_2$ in a planetary ball mill over a period of 60 min, an LE of over 90% was achieved for both components at relatively mild leaching parameters (room temperature (RT), 1 M $H_2SO_4$, 2 vol% $H_2O_2$). Similar positive effects were demonstrated in an experimental study by Yang et al. [46] for the recycling of LFP batteries, where the cathode active material was fine ground in the presence of a chelating agent in a planetary ball mill. The positive influence of mechanical stress on the LE was attributed by the authors to changes in crystal structure. Other approaches to the processing of recycling intermediates are increasingly looking at the use of mechanochemical reactions in grinding media mills. For example, cathode active materials are processed together with reducing agents to catalyze a reaction of lithium metal oxides to lithium aluminate [47]. However, corresponding process steps are not considered in the present study.

### 2.6. Froth Flotation

Froth flotation is a conventional but effective particle separation method currently applied in many industries, such as mineral processing. It relies on the wettability difference between the particle surfaces to achieve separation and enhances the difference in wettability between target and non-target particles by introducing various reagents [48,49]. Flotation is considered to be a promising method applied in Li-ion battery recycling, capable of separating graphite—the anode active material—from the cathode active material. This is mainly due to the significant hydrophobic differences between the natural graphite surface and the particle surface of the cathode active material [30]. In current industrial practice, the leaching of NMC is mainly carried out by acid leaching [17]. After filtration,

graphite can be obtained as a by-product. However, this approach significantly increases the processing volume of the hydrometallurgical process [17]. Therefore, it could help to optimize this industrial process if a flotation process could be added prior to acid leaching. However, in previous studies, it was found that the surface of the cathode active material was covered with binder residues, which reduced the difference in wettability between the surface of the anode and the cathode active material particles, thus reducing the flotation efficiency [30,50,51].

Therefore, proper pre-treatment is required to remove the binder residue prior to flotation. Pre-treatment methods include thermal pre-treatment, mechanical pre-treatment and chemical pre-treatment. Thermal pre-treatment refers to roasting [52] and pyrolysis [53–56]; mechanical pre-treatment contains grinding, cryogenic–grinding [57] and attrition [28]; and chemical pre-treatment mainly refers to the advanced oxidation processes [50,58].

Our previous study compared the flotation results of BM without pre-treatment with chemical pre-treatment and roasting pre-treatment. The best results were obtained with roasting pre-treatment combined with flotation. Recoveries of up to approximately 75% for graphite and 90% for the active material, NMC, were achieved [51]. Vanderbruggen et al. compared the effect of mechanical pre-treatment, thermal pre-treatment combined with mechanical pre-treatment, and electrohydraulic fragmentation on flotation. Thermal pre-treatment combined with mechanical pre-treatment was able to achieve 94.4% graphite recovery and 89.4% recovery of the cathode active material [59]. In addition, attrition-assisted flotation was able to achieve approximately 85% graphite recovery and 70% recovery of the anode active material [28].

Previous studies have more focused on single-stage flotation, and there has not been much research into the effects caused by multi-stage flotation. In this study, multi-stage flotation was introduced to investigate its effect on flotation products.

*2.7. Leaching*

Hydrometallurgy is an essential method of extracting metals, which is extensively used in mineral extractive metallurgy and is widely used in battery recycling today. Leaching is the key step of transferring metals from an ore or raw material into a solution [60,61].

The primary concern in hydrometallurgical treatment for the HP route is the valorization of slags, for example, to extract Li effectively from Li-bearing slags. Elwert et al. investigated the feasibility of leaching Li from Li slags and achieved Li LE of 80...95% when sulfuric acid was used as the leaching agent [61]. Klimko et al. pre-treated Li-containing slags with concentrated acid dry digestion prior to leaching and achieved the leaching efficiency of around 92% [24].

In the case of the intermediate product, BM, being obtained from the WM route and CM route, leaching enables a leachate of Li, Co, Ni, Mn and other metal ions to be obtained. Various leaching methods have been studied for BM, including acid leaching, ammonia leaching and microbial leaching. Inorganic acids like sulfuric acid [62], hydrochloric acid [63] and nitric acid [64], as well as organic acids such as citric acid [65], oxalate [66] and formic acid [67], have been used as leaching agents in acid leaching. During the acid leaching process, reductants such as hydrogen peroxide, ascorbic acid [68] and sodium thiosulfate [69] are often added to increase the LE. The LEs of cathode active materials in different leaching environments are listed in Table 2 below.

**Table 2.** Common leaching agents and LEs of cathode active materials.

| LIB Composition | Leaching Agent | Conditions | Leaching Efficiency |
|---|---|---|---|
| $LiCoO_2$ | 3 M $H_2SO_4$ | 70 °C, 6 h, 20 g/L | Li 80%; Co 97% [62] |
| $LiCoO_2$ | 3 M HCl | 95 °C, 3 h, 100 g/L | Li 99%; Co 100% [63] |
| $LiNi_xMn_yCo_zO$ | 4 M $H_2SO_4$ and 10 mL 50 wt.% $H_2O_2$ | 65–70 °C, 2 h, | Co 94%, Ni 96%, Mn 91% [70] |
| Mixture of $LiCoO_2$, $LiMn_2O_4$, $LiCo_{1/3}Ni_{1/3}Mn_{1/3}O_2$ | 0.5 M citric acid and 1.5 vol% $H_2O_2$ | 90 °C, 1 h, 20 g/L | Li, Co, Ni, Mn > 95% [65] |
| $LiCo_{1/3}Ni_{1/3}Mn_{1/3}O_2$ | 2 M formic acid and 6 vol.% $H_2O_2$ | 60 °C, 2 h, 2 M, 50 g/L | Li~100%; Co, Ni, Mn ~85% [67] |
| $LiCo_{1/3}Ni_{1/3}Mn_{1/3}O_2$ | 1 M acetic acid and 3 mL $H_2O_2$ | 70 °C, 1 h, 20 g/L | Li 98%; Co, Ni, Mn 98% [71] |

Current studies have focused on the direct leaching of BM, while leaching for flotation products has been less commonly studied. Therefore, in this study, leaching experiments were conducted around the products from flotation.

## 3. Experimental Section

### 3.1. Materials

The experimental material was EoL NMC-622 LIBs (modules), provided and shredded (under $N_2$) by industrial partners. The material was subsampled by iPAT and sent to IME for the high-temperature pyrometallurgical test and pyrolysis test and sent to MEET for the electrolyte extraction test. The slag produced by IME was then sent to IFAD for the leaching test. The rest of the experimental material was firstly mechanically pre-treated by MVTAT with sieving, and the fine fraction <0.25 mm was subjected to flotation tests.

### 3.2. Hot Pyrometallurgical Route

3.2.1. Smelting Process

The smelting trials were carried out in a laboratory-scale electric-arc furnace (DC) in a 2 L graphite crucible. The furnace was equipped with a movable, graphite top electrode. Before the feeding of the material, the furnace was preheated to roughly 1000 °C. Afterwards the NMC-622 battery shredder was fed together with fluxes and CuO discontinuously. As fluxing agents, $SiO_2$ (>98%, Quartzwerke GmbH, Frechen, Germany) and CaO (>94.5%, Rheinkalk GmbH, Wülfrath, Germany) were used, based on previous FactSage$^{TM}$ 8.0 calculations. The CuO (>98.9%, Lomberg GmbH, Oberhausen, Germany) was added as an oxidizing agent for the contained graphite in the LIB material. Per trial, 3 kg of LIB shredder and 4.5–6.5 kg CuO were used and heated up to 1600 °C in 120–180° min. A holding time of 10–15 min was carried out after the complete melting of the input material. All trials were carried out in duplicate or triplicate. The procedure is also described by Stallmeister et al. [72].

3.2.2. Slag Leaching

The effect of individual leaching factors such as sulfuric acid concentration, hydrochloric acid concentration, leaching temperature and liquid-to-solid ratio (L/S ratio) on the Li leaching efficiency was investigated at a leaching time of 60 min. Each leaching experiment was conducted under 400 RPM in a beaker (Table 3).

**Table 3.** Leaching parameters.

| Leaching Parameters | | | | |
|---|---|---|---|---|
| Temperature, °C | RT | 40 | 60 | 80 |
| L/S ratio | 15 | 20 | 25 | 30 |
| $H_2SO_4$ concentration, mol/L | 1 | 1.5 | 2 | 3 |
| HCl concentration, mol/L | 1 | 1.5 | 2 | 3 |

In order to investigate the effect of leaching temperature on Li leaching efficiency, the L/S ratio was set at 25, with a sulfuric acid concentration of 2 M. When investigating the effect of the L/S ratio on Li leaching efficiency, the leaching temperature was set at 40 °C and the sulfuric acid concentration was 2 M. When examining the effect of sulfuric acid concentration on Li leaching efficiency, the temperature was set at 40 °C and the L/S ratio was 25. In order to investigate the effect of hydrochloric acid concentration on the Li leaching efficiency, the leaching temperature was maintained at 40 °C and the L/S ratio was set at 25.

3.2.3. The Impact of Ultra-Fine Grinding of Slags on Leaching Efficiency

As part of the leaching experiments, the effect of fine grinding as a mechanical pre-treatment of the slag on leaching behavior and leaching kinetics was investigated. The aim

of the fine grinding was to liberate Li-containing phases, mainly micrometer-sized lithium aluminate crystals, simultaneously increasing the specific surface area of the particulate system and altering the slag microstructure, as has been observed previously [73]. For this purpose, a stirred media mill from Netzsch (LabStar MicrosSerie, Netzsch, Selb, Germany) with zirconia grinding media (d = 850 μm) was used and operated in circuit mode at a stirred tip speed of 10 m/s. Ethanol was used as a suspending medium, and the solids concentration was set to 10 wt.%. The leaching tests were carried out analogously to the leaching experiments described in Section 3.2.2.

### 3.3. Warm Mechanical Route

#### 3.3.1. Pyrolysis

The pyrolysis trials were carried out with the provided LIB shredder in a 20 L gas-tight steel reactor, placed in a resistance-heated furnace. First, 200 g of shredder was placed in an alumina crucible and treated under inert Ar atmosphere with a continuous Ar flow of 14 L/min and a heating ramp of 300 °C/h up to max. 630 °C. Afterwards, a holding time of 40 min was used to ensure a homogenous temperature profile and sufficient reaction time. The off-gas treatment, consisting of a two stage-scrubber system (1. NaOH solution, 2. $H_2O$) and a thermal post combustion, ensured the cleaning of the produced off-gases during the process. To avoid oxidation reactions, the material remained in the inert furnace until RT was reached. Subsequently, the shredder was sieved to <500 μm and a water-leaching step was carried out to recover and quantify the produced $Li_2CO_3$ content in the BM. A total of 20 g of BM was leached in 500 mL of deionized water for 90 min, followed by the filtration and boiling of the solution to recover the dissolved Li salt. The solid products were analyzed by XRD, and elemental characterization was carried out by ICP-OES, the combustion method and an ion-selective electrode. The whole procedure is also described in detail for a previous parameter study in [26].

#### 3.3.2. Flotation with Roasted BM and Leaching

The BM used for flotation was thermally pre-treated by roasting at IFAD in a muffle furnace. The multistage flotation experiments were conducted in a Denver-type flotation machine in an IFAD construction, as shown in Figure 2. A 1 L flotation cell was used for the rougher and scavenger flotation stage. A 0.5 L flotation cell was used for the cleaner flotation stage. ShellSol<sup>TM</sup>D100 (Shell, London, UK) was used as the collector, and MIBC (Sigma-Aldrich, St. Louis, MO, USA) was used as the frother. The froth product and pulp product obtained by multi-stage flotation were separately used for leaching.

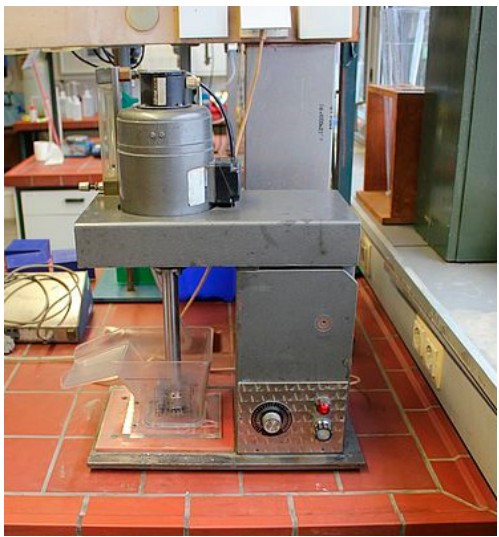

**Figure 2.** IFAD-construction Denver-type flotation machine (IA 0583/3).

Acid leaching tests with 0.5 M and 2 M sulfuric acid at 900 rpm, a leaching temperature of 60 °C and an L/S ratio of 10 were carried out on the froth product mixture and the pulp product mixture.

### 3.4. Cold Mechanical Route

3.4.1. Sorting Process

The pouch cells described in Section 3.1 were crushed and provided by an industrial partner on an industrial scale using a rotor shear and a 10 mm discharge grate. The crushing energy introduced was not measured but was in the range of <8 kWh/t [74]. After inert comminution, the material was dried to separate the volatile solvents.

In order to separate the dry material into different product concentrates, a combination of different comminution, classification and sorting processes was investigated. The basis of the investigation was the flow diagram tested at TUBAF. The sorting units used were a zigzag classifier (TUBAF type) and an air shaking table (Co. TRENNSO-TECHNIK, Weißenhorn, Germany). For the de-coating and reshaping of the electrode foils, a fine impact mill (Co. Hosokawa Alpine) was investigated in addition to the micro vortex mill (Co. Görgens, Dormagen, Germany) currently used at TUBAF.

3.4.2. Electrolyte Extraction

For the comparability of recovery rates, in-house-made NMC622||graphite 18,650 cells were built and filled with 4000 mg of a typical LIB electrolyte (1 M $LiPF_6$ in EC/EMC (3/7) + 2 wt% VC) The cells were solely filled and wetted (20 h) for homogenous electrolyte distribution but did not undergo any cell formation. In contrast to aged cells or even unknown and inhomogeneous shredded material, the usage of well-defined material ensured sample comparability to study the co-solvent's influence on carbonate and salt recovery.

$CO_2$ extractions were performed based on previous reports from Grützke et al. [43] under subcritical conditions (40 °C, 60 bar) with varying co-solvents. The overall extraction duration was reduced by 73% from 165 min to 45 min, including 5 min of static equilibration under $CO_2$, 3 co-solvent steps of 5 min of flow-through, 5 min of static equilibration time and finally, 5 min of $CO_2$ flow. Additionally performed elongated extractions resulted in comparable extraction rates (>85%) for the 3ACN/PC system as reported by Grützke et al. [43] The 3 most promising co-solvents were compared in more detail: the mixture of ACN and PC (3/1), acetone and ethyl acetate.

The obtained extracts were weighed and carbonates as well as the conducting salt were quantified using gas chromatography–flame ionization detection (GC-FID) and ion chromatography–conductivity detection (IC-CD), respectively.

## 4. Results and Discussion

### 4.1. Hot Pyrometallurgical Route

4.1.1. Smelting Trials

Different slag systems were investigated in the frame of the smelting trials regarding the recovery yields of Ni, Co and Cu in the metal phase and Li slagging, as already published in [72]. The ratio of $SiO_2$:CaO and the total flux amount addition (350 g/1 kg shredder and 450 g/1 kg shredder) were varied based on previous thermochemical calculations. In the trials, it was shown that the best results were achieved by the addition of 350 g fluxes per 1 kg of shredder with a ratio of 40:60% $SiO_2$:CaO. The best achieved metal yields and the Li slagging rates are summarized in Table 4. Therefore, the following process steps were carried out with the slag produced from these trials.

**Table 4.** Best achieved metal yields and Li slagging rates in smelting trials.

|  | Cu | Ni | Co | Mn | Li |
|---|---|---|---|---|---|
| Metal yield/Li slagging [%] | 99.8 | 99.6 | 98.1 | 56.1 | 71.9 |

### 4.1.2. Leaching of Slags

Based on Table 5, the slag had a high concentration of Al, Ca and Si. Furthermore, it contained up to 2.59% Li, which is nearly comparable to the Li content of a spodumene concentrate.

**Table 5.** Chemical composition of slag sample.

| Al | Ca | Co | Cr | Cu | Fe | Li | Mg | Mn | Na | Ni | P | Si | Zn | F |
|---|---|---|---|---|---|---|---|---|---|---|---|---|---|---|
| % | % | % | ppm | % | % | % | % | % | ppm | ppm | % | % | ppm | % |
| 17.8 | 16.9 | 0.13 | 289 | 0.81 | 0.34 | 2.59 | 0.22 | 2.48 | 66 | 778 | 0.21 | 7.57 | 117 | 0.46 |

When the slag was leached using 1 M sulfuric acid, the LE of Li reached approximately 98% (Figure 3), and when the acid concentration was further increased, then the LE could reach approximately 100%. In contrast, the LE was slightly lower than that of sulfuric acid at the same concentration of hydrochloric acid. Temperature had a significant effect on the LE. Leaching at ambient temperature using sulfuric acid resulted in an LE of approximately 74%, while increasing the leaching temperature to 40 °C resulted in an LE of approximately 96%. When the temperature was further increased, the LE approached 100%. In addition, no significant trend was observed when the liquid–solid ratio was changed from 15 to 30, and the LE ranged from 88% to 94%.

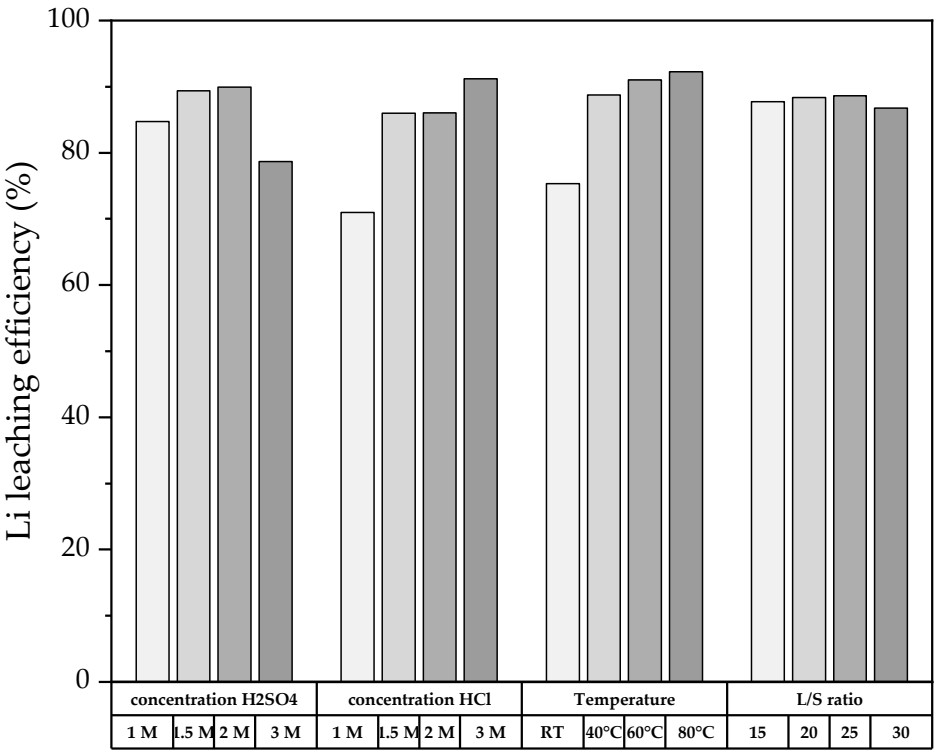

**Figure 3.** Leaching efficiency of lithium over a 60-min leaching period.

The results above reveal the feasibility of the direct leaching of Li-bearing slag. The LE of Li was close to 100% under suitable leaching conditions, which is compatible with previous studies [24]. However, in some experiments, the appearance of silica gel could be observed in the leachate. The occurrence of silica gel is related to the concentration of silicon in the leachate. As silica gel is unfriendly to the hydrometallurgical process and can hinder filtration, desilication through flotation appears to be a viable treatment method for slag to address this issue. If silicate minerals can be separated from the slag before leaching, this will considerably decrease the amount of silica in the leachate.

The mineralogical study of the slag showed that Li was present in the $LiAlO_2$ phase in the slag (refer to Figure 4), while the gangue minerals in the slag were mainly calcium silicate minerals such as melilite solid solution. According to the previous experimental study of micro-flotation by our research group, it was able to achieve a $LiAlO_2$ yield of higher than 65% under the use of fatty acid-based collectors such as sodium oleate [49]. Therefore, slag flotation will be further investigated in a subsequent study.

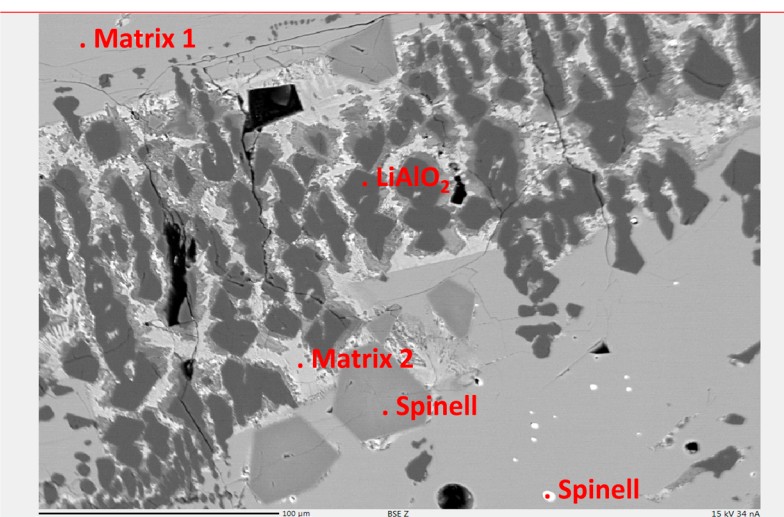

**Figure 4.** Mineralogical composition of the slag sample.

### 4.1.3. Fine Grinding of Slags and Its Effect on Leaching

Based on the results of direct slag leaching, the effects of an upstream fine-grinding-process step on the LE and leaching kinetics were investigated. Three milled samples with a specific surface area $S_m$ of 0.2 $m^2$/g, 4.5 $m^2$/g and 16 $m^2$/g were used for leaching tests, whereby only the latter sample was produced using a fine-grinding process in a stirred media mill. The other two samples ($S_m$ = 0.2 $m^2$/g and $S_m$ = 4.5 $m^2$/g) resulted from dry pre-grinding-process steps. The LE was evaluated over process time as a function of specific surface area, leaching temperature, and acid concentration, using only 1 M and 2 M sulfuric acid with a liquid–solid ratio of 25. Based on the initial results of direct leaching, temperatures of 20 °C and 40 °C were investigated. The objectives of the fine grinding were the liberation of Li-containing phases increasing the specific surface area and altering the slag microstructure in order to enhance the leaching kinetics and to reduce the required amounts of thermal energy and leaching reagents.

The results of the leaching experiments are shown in Figure 5, which illustrates the LE of Li for 1 M and 2 M sulfuric acid, respectively. It demonstrates that an LE of up to 98% was achieved independent of the acid concentration, requiring a temperature of 40 °C and a specific surface area of 16 $m^2$/g using 1 M sulfuric acid and 4.5 $m^2$/g using 2 M sulfuric acid. Moreover, it is easily seen that the LE was lower at a temperature of 20 °C, independent of the acid concentration and the specific surface area of the slag.

When 1 M sulfuric acid was used, a clear dependence of the LE and typically of the leaching kinetics on the specific surface area emerged, which was due to the liberation of the Li-containing phases with reduced particle size and in particular to the increased reactivity of the particulate system. Overall, the LE for a two-hour leaching period at T = 40 °C was increased from 84.8% ($S_m$ = 0.2 $m^2$/g) to 97.0% ($S_m$ = 16 $m^2$/g). In contrast, the use of 2 M sulfuric acid as a leaching reagent showed that the specific surface area of the slag at a temperature of 40 °C only positively influenced the leaching kinetics but did not significantly affect the LE at a process time of two hours. Here, fine grinding offered the possibility of reducing the leaching time to 30 min to achieve a relevant LE of at least 90%.

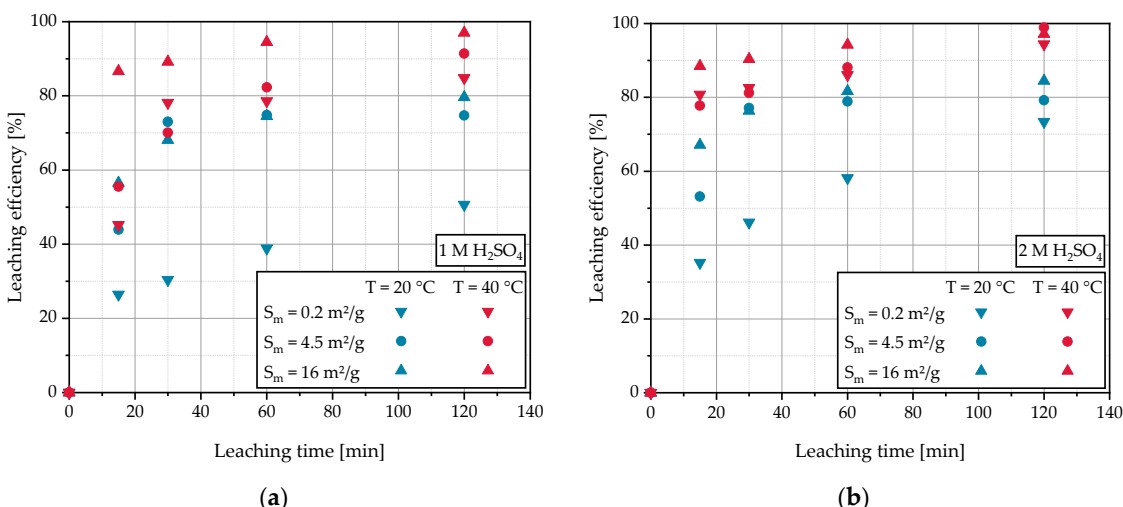

**Figure 5.** Leaching efficiency of fine-ground slags using 1 M $H_2SO_4$ (**a**) and 2 M $H_2SO_4$ (**b**).

### 4.2. Warm Mechanical Route

#### 4.2.1. Thermal Treatment and Water Leaching for Li Recovery

In our previously published study, the detailed investigation of the influence of holding time and temperature during the pyrolysis of the LIB shredder under Ar atmosphere was carried out [26]. It was shown that the pyrolysis temperature had the main effect on the generated products. With a rising temperature, the mass loss of the sample rose as well, due to enhanced organic removal. After a treatment above 505 °C, no further organic residues were detectable in the sample. The removal of binders from the active material was particularly beneficial for the following separation of BM from collector foils, shown by a rising fine fraction share <500 μm after sieving up to 639 °C. Accordingly, an organic and fluorine-containing off-gas, mainly consisting of electrolyte components, HF, hydrocarbons, $CO_2$ and CO was produced during the thermal treatment, leading to reducing reactions of the cathodic metal oxides, as shown in Figure 6.

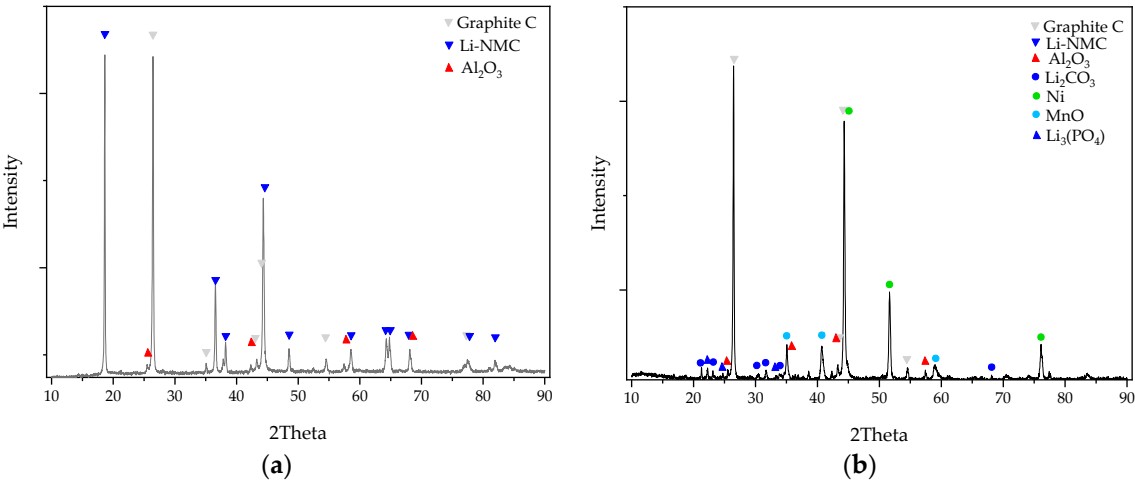

**Figure 6.** XRD analysis of untreated black mass (**a**) and black mass pyrolyzed at 630 °C (**b**).

The reducing atmosphere in combination with elevated temperatures resulted in the formation of metallic Ni and single oxides of Co and Mn. As reported in the previously published study by the authors [26], these reactions lead to the liberation of Li and its phase transformation to $Li_2CO_3$. By means of water leaching of the BM, Li was selectively extracted from the BM with a total yield of 62.4% after a thermal treatment at

642 °C. Lower temperatures, on the other hand, led to lower Li yields due to incomplete reduction reactions.

Further analysis of the Li-salt product showed that it mainly consisted of $Li_2CO_3$. The main impurities were identified to be F and Al, originating from the co-leaching of LiF and Al dissolution due to the basic pH value (~10–11) of the $Li_2CO_3$-containing solution.

### 4.2.2. Froth Flotation of Black Mass

In our previous experimental flotation studies performed in a small Denver flotation machine (125 mL cell), graphite recovery was up to about 75% and graphite content in the froth product could achieve 77% [51]. In this study, the roasting time was first compared (Figure 7a). When the roasting time reached 60 min, the graphite content in the froth product remained stable at 64–65%, irrespective of whether it was a single flotation stage or an additional cleaner flotation stage. However, when the roasting time extended to 90 min, a significant increase in graphite content in the froth product was observed, regardless of whether it was from rougher or cleaner flotation. The froth product derived from the one-stage flotation process exhibited 89% graphite content, while the inclusion of a cleaner flotation stage led to a slight increase, bringing the graphite content in the froth product to 93%. From the flotation results above, the purity of the graphite in the froth product was improved after a suitable roasting time. This shows that roasting plays a decisive role in the subsequent flotation.

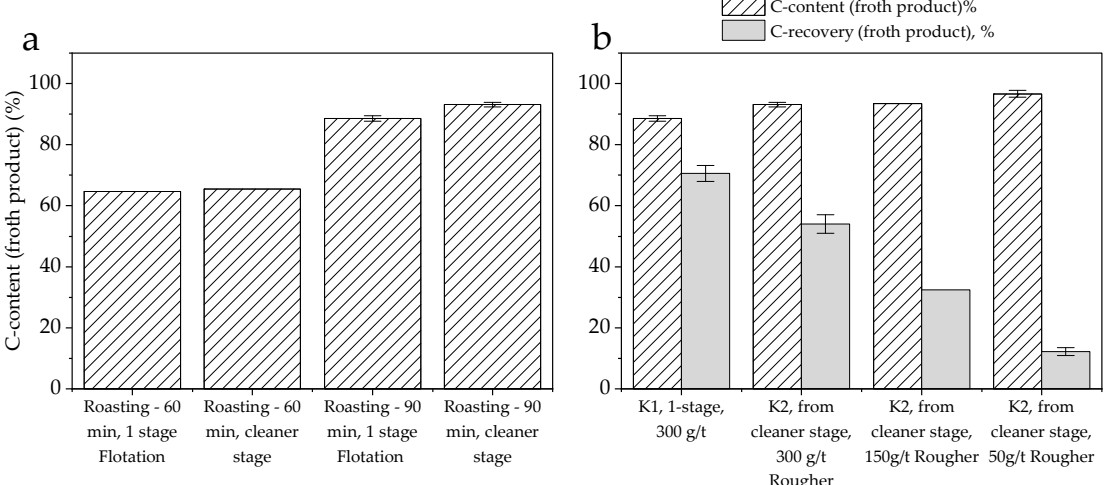

**Figure 7.** (**a**) C-content in the froth product at different holding times and stages at 450 °C; (**b**) C-content and recovery in the froth product at different collector dosages.

In addition, Figure 7b shows that the graphite content in the froth product continued to increase with decreasing collector dosage in the rougher stage. As the collector dosage was reduced to 50 g/t, the graphite content in the froth product increased to approximately 96%. However, this decrease in collector dosage also resulted in a decline in graphite recovery from about 54% to only 12% during this stage.

Pulp product is a concentrate enriched with the cathode active material, NMC. The Ni content in the pulp product increased to 28% (about 75% NMC content) after one-stage scavenger flotation, with a recovery of up to 96%. After two-stage scavenger flotation, the Ni content in the pulp product reached approximately 30% (about 80% NMC content), with a recovery of up to 86%.

### 4.2.3. Leaching and Refining Graphite

The graphite concentrate obtained from the multi-stage flotation process required the further removal of impurities to increase the graphite content. The froth products, K2, obtained from the various multi-stage flotation processes were mixed and probed. The pulp product, B3, was also mixed and probed. The elemental analyses are given in Table 6.

**Table 6.** Chemical composition of the K mixture and B mixture.

| Sample | Al | Co | Cu | Li | Mn | Ni | C |
|---|---|---|---|---|---|---|---|
| | % | % | % | % | % | % | % |
| K Mixture | 1.8 | 0.6 | 0.5 | 0.6 | 0.6 | 1.8 | 90.6 |
| B Mixture | 4.0 | 9.9 | 1.6 | 5.0 | 9.5 | 30.0 | 5.9 |

Acid leaching was then carried out separately on the K mixture and B mixture. According to the results, the graphite content of the K mixture was further increased after acid leaching (Figure 8a). After 0.5 M sulfuric acid leaching, the carbon content in the graphite concentrate was 97.2%. The carbon content in the graphite concentrate after 2 M sulfuric acid leaching was 98.4%. Elemental analysis of the graphite product after acid leaching showed that Al was a major impurity in the graphite concentrate (Figure 8b).

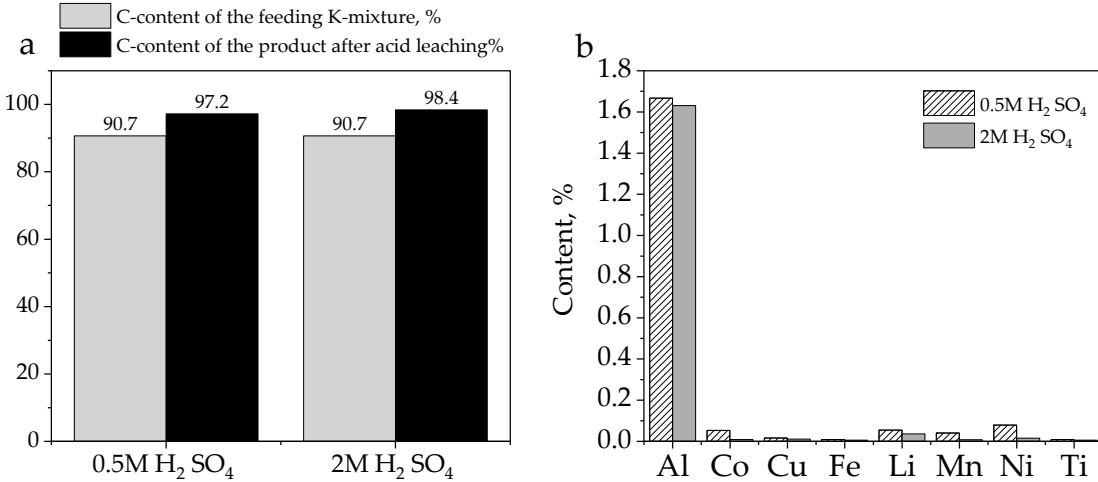

**Figure 8.** (**a**) Graphite content before and after leaching with 1 M and 2 M sulfuric acid; (**b**) main metal content of the graphite product after leaching.

After the leaching tests with the B mixture, the LE was only about 35% for Co and 30% for Ni at a sulfuric acid concentration of 0.5 M (Figure 9). At a sulfuric acid concentration of 2 M, the LE of Co increased to 77%, and that of Ni increased to 85%.

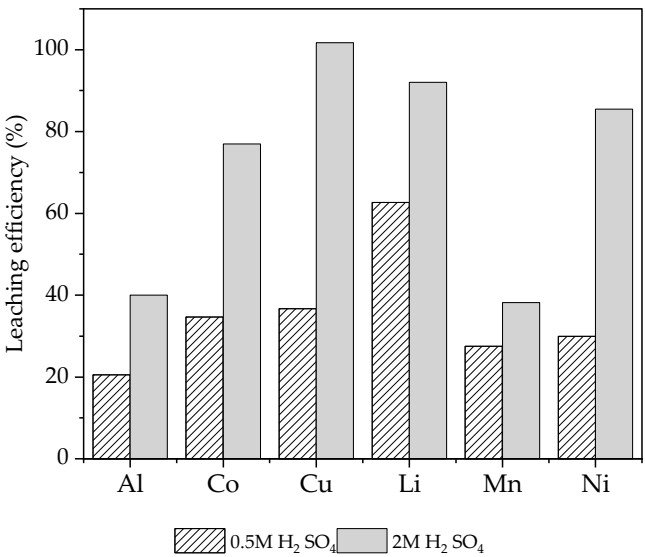

**Figure 9.** Leaching efficiency of main metals.

*4.3. Cold Mechanical Route*

4.3.1. Electrolyte Extraction

The obtained shredded feedstock material already underwent thermal treatment to regain significant amounts of low-boiling-point compounds. Nevertheless, as investigated comprehensively in a previous study, significant residues of organic carbonates, additives and decomposition species were present [75]. These compounds were extractable from the headspace above the shred or by using solvent extraction by dichloromethane and were analyzed by chromatographic techniques coupled to mass spectrometry. In addition to these organic residues, the conducting salt was not recovered by evaporation and therefore was also extractable from the shredded feedstock.

To extend the electrolyte recovery beyond the evaporation step, (solvent-assisted) $CO_2$ extraction was applied. The focus was the improvement of salt recovery; however, carbonate recoveries were also monitored to illustrate $CO_2$ extraction capabilities for processes without an initial thermal evaporation step. The literature-reported method was adjusted to focus on options for faster (45 instead of 165 min) and more environmentally friendly and cost-efficient solvent-assisted $CO_2$ extraction [42,43] (Table 7).

**Table 7.** Studied co-solvents with their net prices for ordering 1 l from Sigma-Aldrich (31 January 2023) and maximale Arbeitsplatz Konzentration (MAK-Wert) according to GESTIS Stoffdatenbank of the German Institut für Arbeitsschutz.

| Solvent | Price/EUR L$^{-1}$ | Rel. Price Diff. | MAK/mg m$^{-3}$ |
|---|---|---|---|
| ACN/PC (3:1) | 112.68 ($\geq$99.5%) * | $\pm$0% | 17 (ACN), 8.5 (PC) |
| Acetone | 57.20 ($\geq$99.5%) | $-$49.2% | 1200 |
| Ethyl acetate | 50.10 ($\geq$99.5%) | $-$55.5% | 750 |

* is the reference value (100%) for the later price calculations.

The obtained recovery rates for the three considered co-solvents are given in Figure 10.

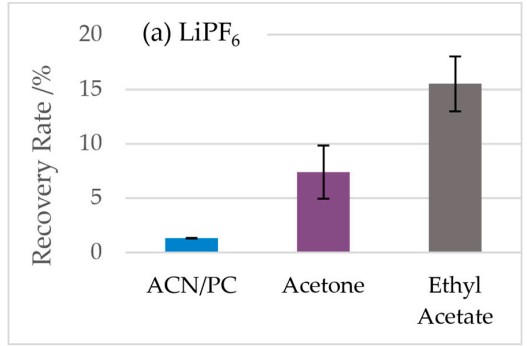
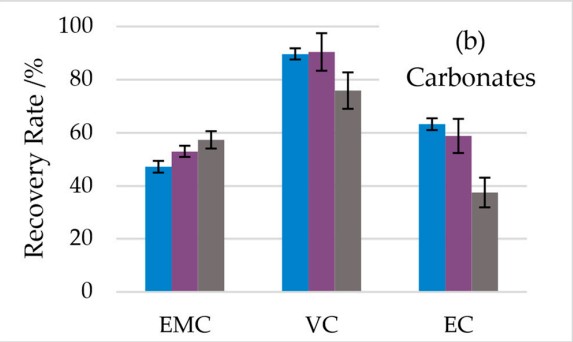

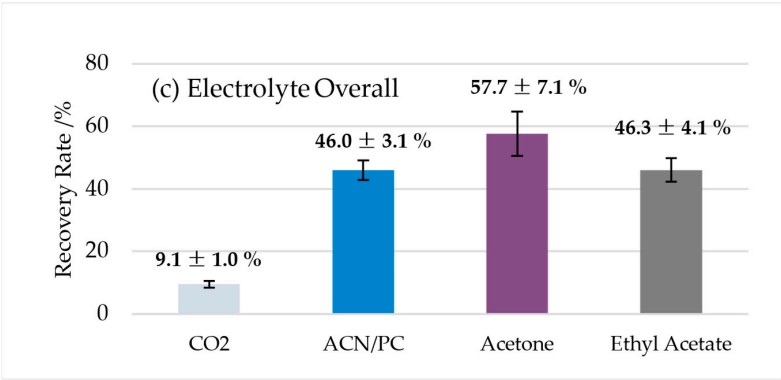

**Figure 10.** Obtained recovery rates of LiPF$_{6s}$ (**a**), the used carbonates (**b**) and the electrolyte overall (**c**).

While pure $CO_2$ did not extract any $LiPF_6$ within the short extraction duration, the extracted amounts for acetone (7.4%) and ethyl acetate (15.5%) were significantly higher than those for the 3ACN/PC (1.3%) experiments. The low overall extractions rates were caused by the significantly lower extraction duration. Nevertheless, remarkable improvement was obtained for the conduction of salt extraction using a faster method with more economic and ecological co-solvents compared to literature reports [42,43].

The extraction of carbonates, especially the cyclic carbonates, was more efficient, while the comparison of the co-solvents used showed trends with increased linear carbonate extraction from 3ACN/PC (47.2%) over acetone (53.0%) to ethyl acetate (57.4%), and the cyclic carbonate extractions improved from ethyl acetate with significantly lower extracted amounts of EC (37.5%) over acetone (58.8%) to 3ACN/PC (63.2%). This is in line with the solvent polarity. The more polar co-solvents showed improved cyclic (~polar) carbonate and worse linear carbonate extraction, while less polar co-solvents showed reversed trends. These trends should be respected for carbonate recovery via solvent-assisted $CO_2$ extraction. When thermal evaporation is applied, mainly cyclic carbonates extraction will be important, while without evaporation, removal and regaining the linear carbonate are also of importance and could be tailored via the used co-solvent. All in all, solvent-assisted $CO_2$ extraction is able to remove and regain carbonates effectively, as well as with significantly shortened extraction durations and the newly considered co-solvents.

The handled shredded material was also extracted, and residual carbonates as well as conducting salt were found. This is in line with extensive qualitative analyses and shows that, despite thermal evaporation, electrolyte recovery can be significantly improved using extraction methods [75]. However, the inhomogeneity of the shredded material and its unknown quantitative composition did not allow for a reliable calculation of extraction rates, which was also why previously shown basic comparative studies with different co-solvents were performed with well-defined samples rather than the complex shred.

For application in a "CM route", extraction parameters and the use of co-solvents should be tailored according to the targeted compounds since the investigated solvents showed contrary trends for salt and carbonate removal, and carbonate removal could even be distinguished in terms of the effectiveness of linear or cyclic carbonate extraction depending on the co-solvent's properties. From economic and ecological perspectives, the applicability of more common solvents was demonstrated and resulted in significantly faster conducting-salt recovery.

### 4.3.2. Mechanically Produced Material Concentrates

The liberation of the crushed InnoRec material was determined to be $\geq$99% by the manual sorting of a representative sample, based on the main components (separator, housing, anode, cathode and solid plastic). Thus, the liberation was sufficiently high to separate these components from each other. The extent to which a less-intensive comminution would also have produced complete liberation was not investigated. Under comparable circumstances, discharge grates of 30 mm were also found to be sufficient [74].

As the first step, the fines (x < 0.5 mm), or the so-called BM, were separated by screening.

Figure 11 shows that this fraction constituted a major part of coating material, which could thus be enriched in a product. In addition, it reduced the mass to be further processed by about 50%. However, the results also showed that the particle size class up to 3.15 mm still contained active material that was still adhering to the electrode foils. Due to the additional stress, the electrode foils needed to be further de-coated.

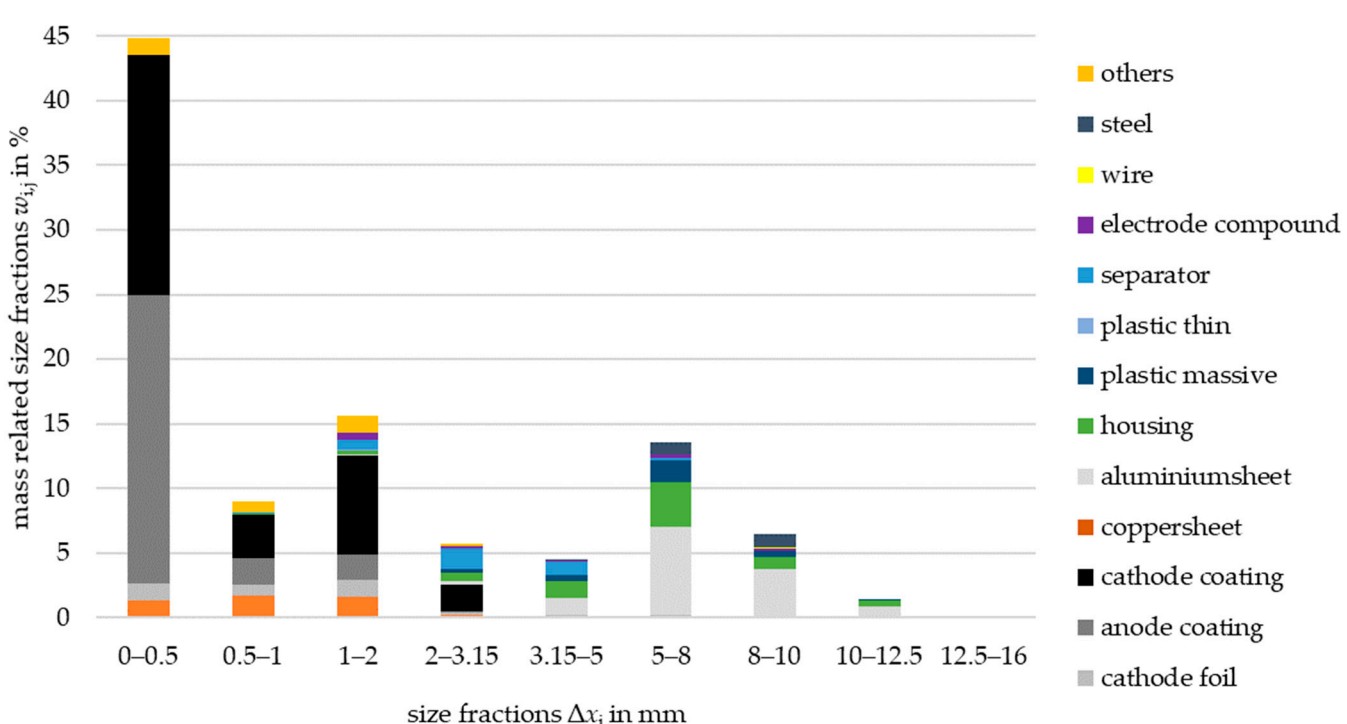

**Figure 11.** Mass-related size fractions of each material.

This was followed by two-stage airflow sorting at air velocities of about 1.8 and 8.8 m/s. The result (refer to Figure 12) was a light product containing the separator foil and a heavy product containing the module and cell housing pieces. The entire scheme of the mechanical processing of the crushed modules is shown in Figure 13. The next two steps were ultimately a repetition and served only to improve the result. Due to the "stress" of the material in the zigzag air classifier, composites were liberated, and the electrode foil was partially de-coated, which was why the repetition was necessary. The result was another BM product and a heavy product with housing parts.

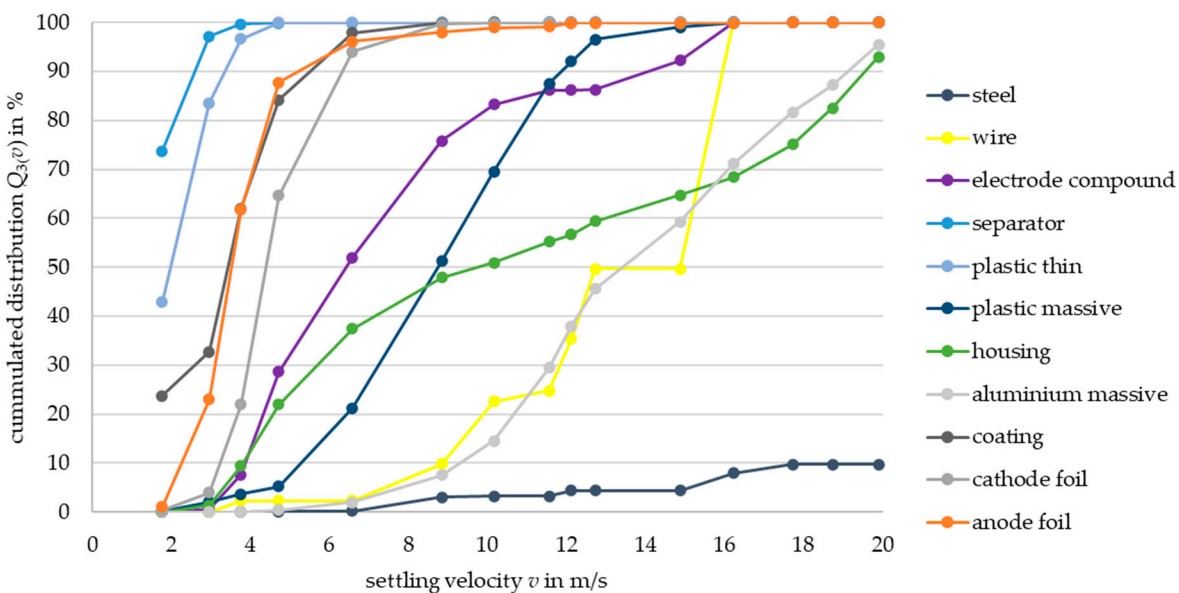

**Figure 12.** Settling velocity distributions.

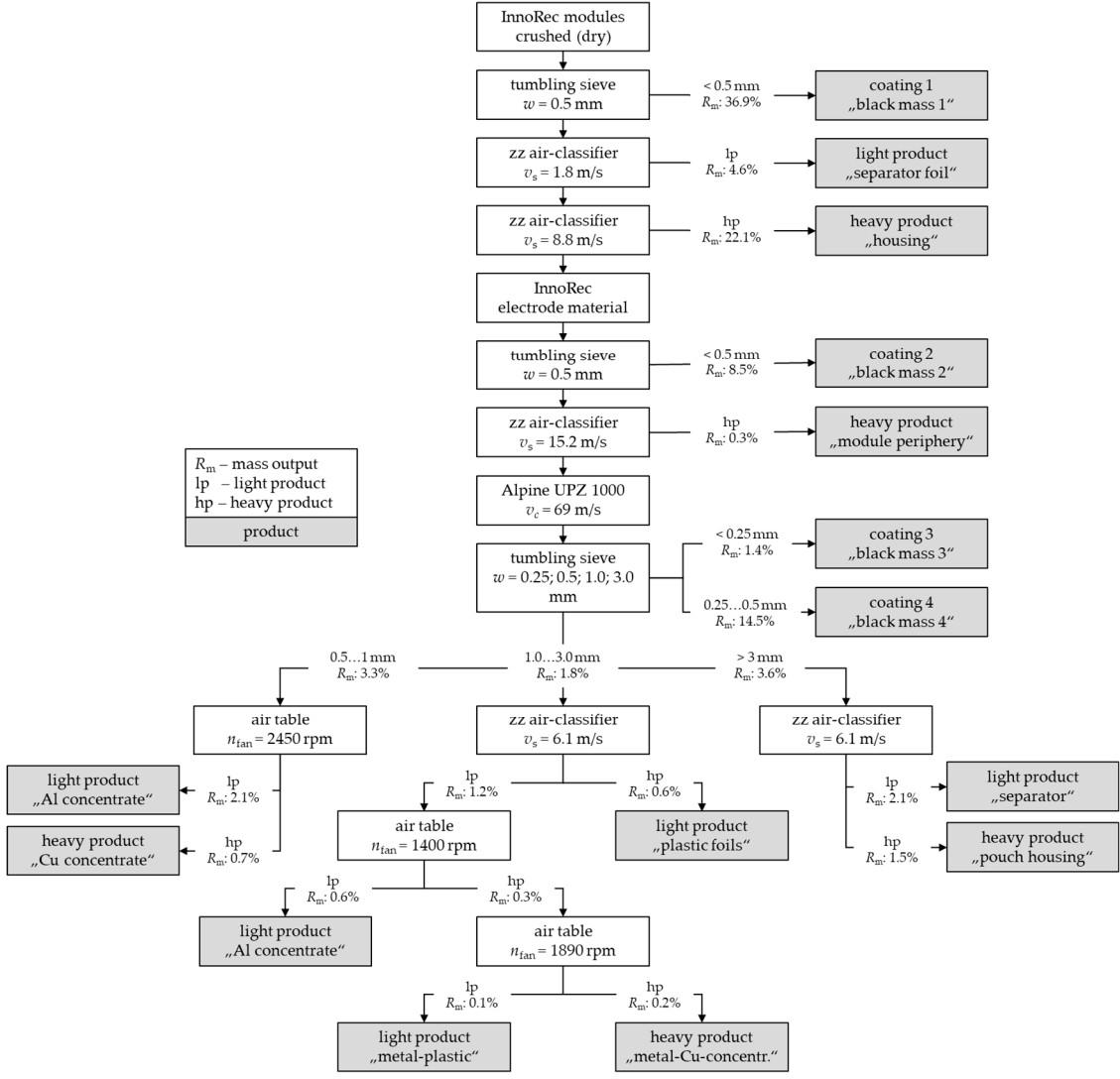

**Figure 13.** Process flow chart of mechanical processing.

What remained was an intermediate product of residual coated electrode foils with a mass fraction of approximately 25%, based on the dry initial weight. These foils had the same particle size and settling velocity distribution. For these reasons, the material was stressed again. This was to make the foils more spherical by reshaping. At the same time, this removed the rather brittle coating from the foils. In this case, a fine impact mill from Hosokawa Alpine was used. After stressing, another BM x < 0.5 mm resp. x < 0.25 mm could be screened (see Figure 13, black mass 3 and 4).

The reshaped, de-coated classified foils that and narrow particle-size ranges were then separated into various concentrates by further sorting processes (cf. Figure 13). The remaining mass fraction due to this material was about 10%. Here, a combination of the already-used zigzag air classifier and an air shaking table was used. The upstream classification and the interconnection of several sorting steps allowed comparatively high purities of the concentrates to be produced. For example, the heavy product in the 0.5…1 mm size class consisted of 85 wt.-% copper, and the light product in the >3 mm class consisted of 98 wt.-% plastic.

This figure also shows all other mass flows into the respective products. The two largest product streams were the BM with more than 60 wt.-% and the housing fraction with more than 20 wt.-%. Here, the BM was composed of ≥90% electrode coatings. The housing fraction contained approx. 80 wt.-% Al and 20 wt.-% steel. This fraction could also be further processed by downstream processes such as magnetic separation.

*4.4. Comparison of the Three Recycling Routes*

From the perspective of recovery, about 75–80% of the Li entered the slag in the HP route. From the slag, an LE of Li > 90% could be achieved. In this route, two main issues needed to be considered, i.e., Li-bearing fly ash needed to be further recycled, which was not possible due to the lab-scale facilities, and the silica gel generated by some of the high-silica slag during the leaching process needed to be dealt with. In contrast, the WM route achieved an Li recovery of about 63% after pyrolysis treatment by water leaching. Since this concept is a quite new and innovative one, its optimization has not been completed yet.

The graphite, however, could not be effectively separated from the cathode active material—NMC—by flotation in the cold mechanical route, mainly because residues such as binders on the surface of BM particles were not effectively removed, thus greatly affecting its separation efficiency. Therefore, if graphite and NMC cathode active materials are to be separated by flotation, organic residues on the BM surface need to be removed by pre-treatment. Though the BM can be leached directly in the CM route, the corrosion of the equipment by halogenic components and the effect of other organic residues on the hydrometallurgical process is a serious problem [1,17]. In the WM route, after pre-treatment by roasting, a graphite yield of about 75% was achieved, with the addition of cleaner-stage purity of up to 96%. After the leaching of the floatation concentrate, the purity of the graphite concentrate could be increased to 97–98%. However, roasting led to a loss of graphite. But if thermal treatment was carried out under optimal conditions, e.g., pyrolysis, there would have been no loss in graphite. The recovery of BM obtained after pyrolysis pre-treatment can reach 86–94% after combining with the attrition process according to Vanderbruggen et al. [28,59], but different pyrolysis parameters likewise produce different pyrolytic residues [54,55], which have an impact on flotation results. Therefore, the thermal treatment conditions of this route need to be further investigated. In contrast, the HP route consumed graphite as a reducing agent in the melting process but required a less complex process chain.

Co and Ni could be recovered at >98% in the HP route in an alloy. In the WM route, after thermal treatment and by adding a scavenger stage in flotation process, Co and Ni could be recovered in the form of NMC with a recovery of about 96%. In this case, the Ni content in the pulp product was 28%. According to Vanderbruggen et al., the recovery of Co in the form of LMO reached 89% after pyrolysis treatment combined with high shear pre-treatment in the flotation stage [28]. In the subsequent hydrometallurgical process, the NMC-rich product was leached. Leaching efficiencies of Co and Ni up to 77–85% could also be achieved by only adding 2 M $H_2SO_4$.

In the HP route, Al was used mainly as a reducing agent, and most of the Al entered the slag as aluminate. In the other two routes, the Al was separated and enriched in the form of a foil fraction. The recovery of Al in the WM route was about 65%, while its recovery in the CM route was more than 87%. Through the thermal treatment, the binder was destroyed, which allowed for the better de-coating of the Cu foils and thus enhanced the separation of the remaining active materials. However, there was also the corrosion of the Al foils, presumably caused by oxidation, which resulted in an increased amount being separated into the fine fraction rather than the electrode fraction.

Among these three recycling routes, the WM route was able to recover the most abundant components from the LIBs. However, this route and the CM route are very sensitive to the composition of the feed. The inhomogeneity of the raw material thus affects the stability of the process. In the CM route, the presence of fluorine, graphite and organic residues can seriously affect the subsequent hydrometallurgical processing. In contrast, the WM route helps to solve this problem [17]. In addition, from the point of view of industrial applications, the process robustness of the HP route makes it suitable for the large-scale treatment of LIB materials and can effectively reduce the processing cost, yet the energy consumption is high [17]. Furthermore, during the smelting process, halogenic and other

organic residues as well as heavy metal dust can be collected by off-gas treatment [17]. But the recovery of Li and graphite requires further investigation and optimization.

Based on the above process, Figures 14–16 show the Sankey diagrams of these three investigated technological recycling routes. The data used here are derived from part of experimental data in the InnoRec project.

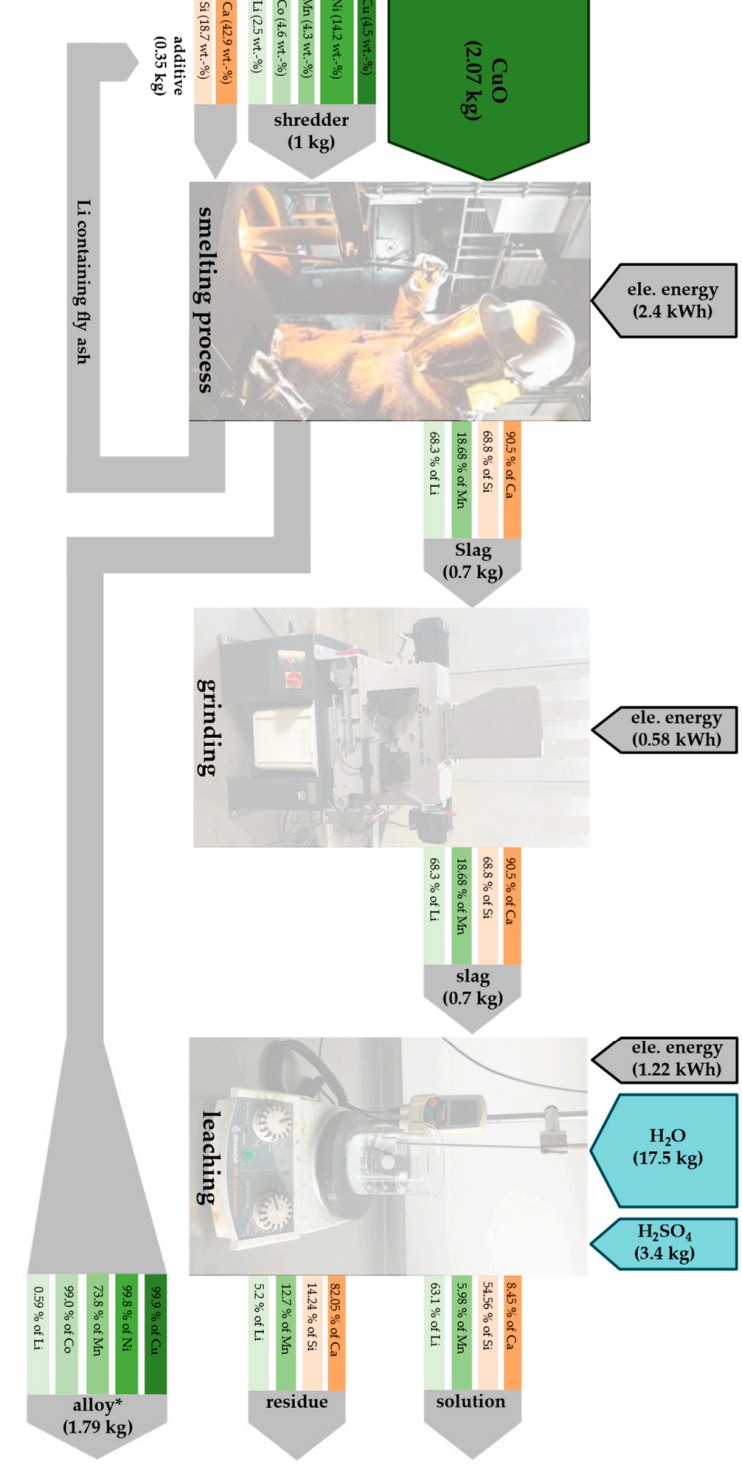

**Figure 14.** Sankey diagram of the HP route.

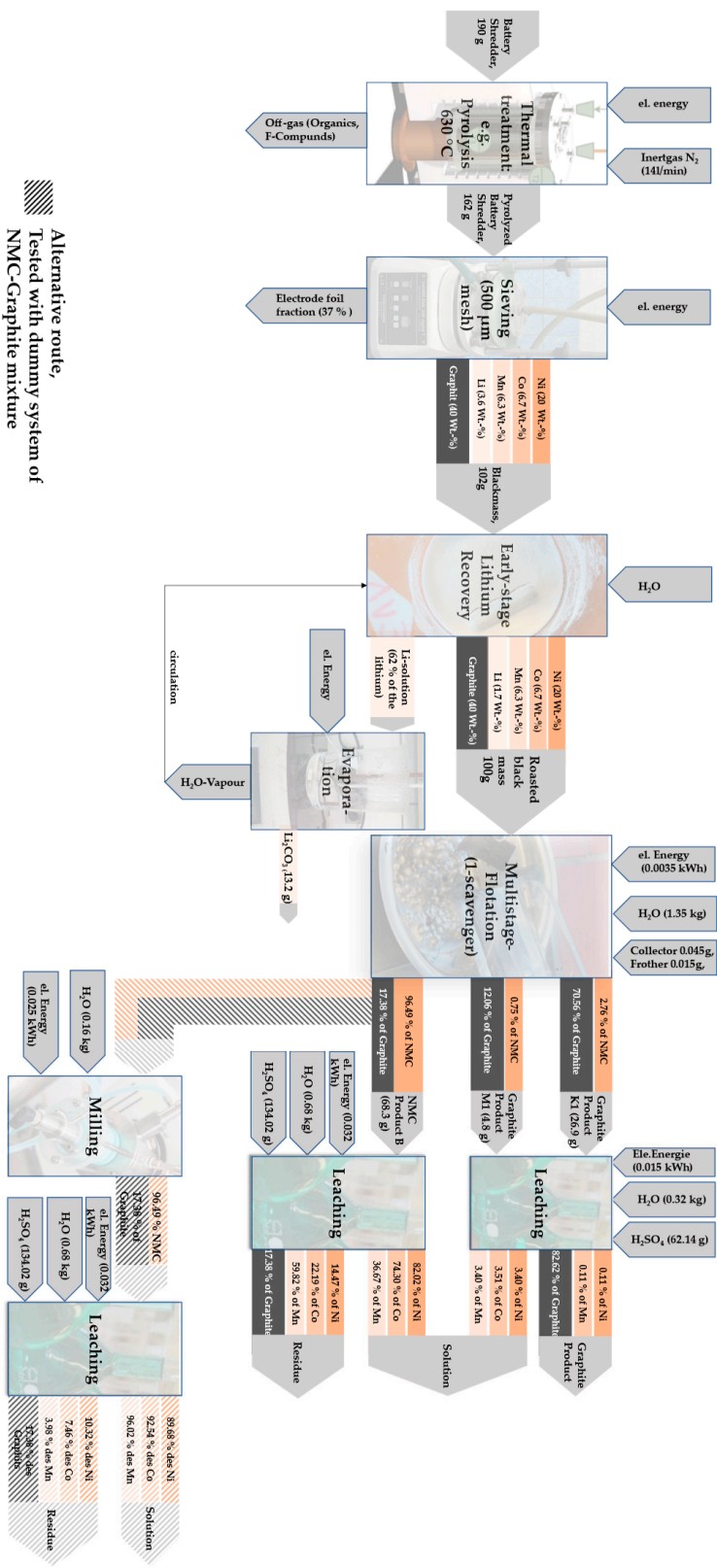

**Figure 15.** Sankey diagram of the WM route.

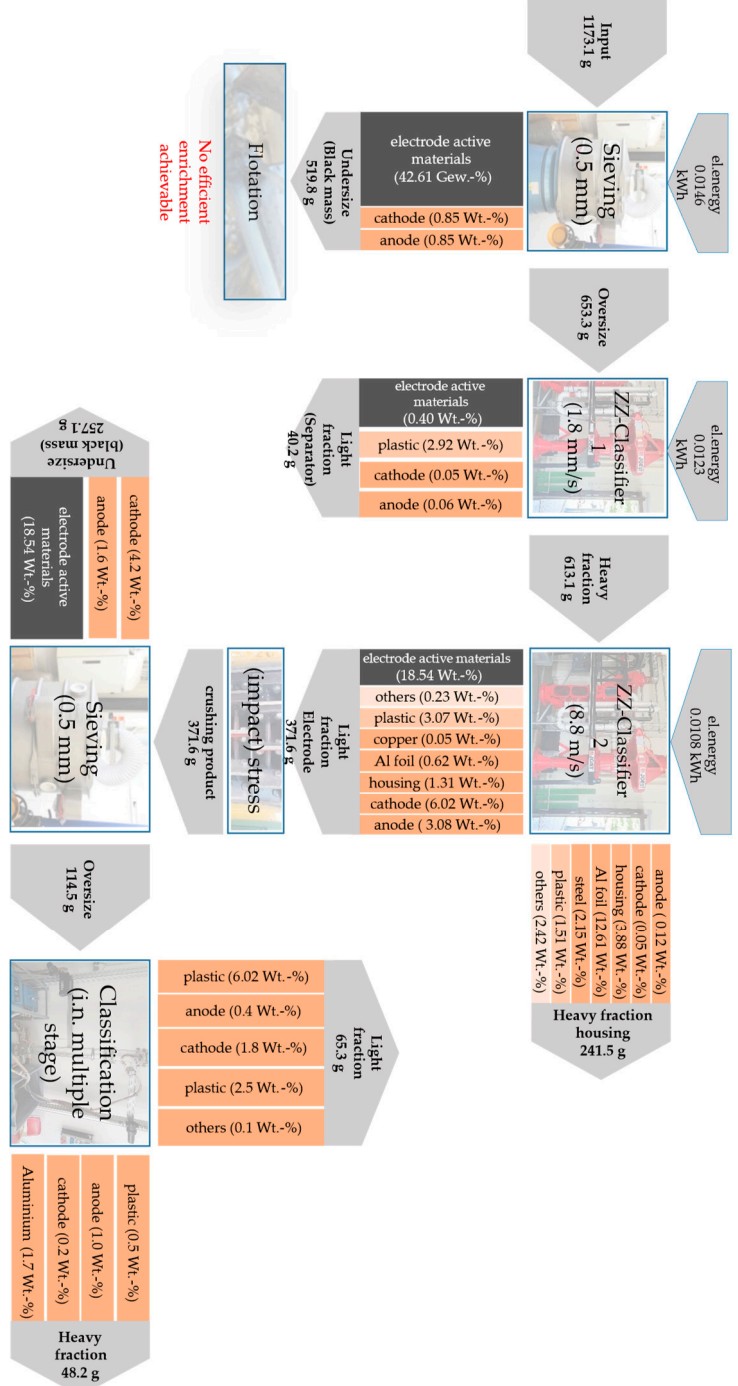

**Figure 16.** Sankey diagram of the CM route.

## 5. Conclusions

This paper compared three representative recycling routes: the hot pyrometallurgical route, the warm mechanical route and the cold mechanical route. In addition, the same feed (LIB module, NMC-622) was used for all three recycling routes in this paper, and six elements (Al, Cu, C, Li, Co and Ni) were selected to compare their recoveries.

The three different recycling routes represent specific application scenarios, each with their own advantages and disadvantages. As the chemical composition of battery materials and various doping elements continue to change today, these three recycling routes should be combined in some way to improve the overall recycling efficiency of batteries. For example, the HP route and WM route can be integrated. Li can be pre-extracted via the

WM route, with the separation of graphite, followed by the recovery of metals such as Co and Ni through the HP route. This study still has some limitations, such as the fact that no pilot-scale experiments were conducted.

The InnoRec project forms the bridge and basis for recycling projects in the greenBatt cluster. The greenBatt competence cluster is dedicated to establishing a closed loop for battery materials and resources, focusing on an energy- and material-efficient battery life cycle. The different projects in the cluster will continue to deepen the research on these three recycling routes and carry out economic and environmental analyses.

**Author Contributions:** Conceptualization, H.Q. and D.G.; methodology, H.Q., C.S., M.T., C.P. and T.L.; investigation, H.Q., C.S., M.T., C.P. and T.L.; resources, D.G., B.F., A.K., M.W., S.N. and U.A.P.; data curation, H.Q., C.S., M.T., C.P. and T.L.; writing—original draft preparation, H.Q., C.S., M.T., C.P., S.N. and T.L.; writing—review and editing, all authors; supervision, D.G., B.F., A.K., M.W. and U.A.P.; All authors have read and agreed to the published version of the manuscript.

**Funding:** This research was funded by the Federal Ministry of Education and Research (BMBF) within the research program: "ProZell 2" (InnoRec, reference number: 03XP0246A, 03XP0246B, 03XP0246C, 03XP0246D and 03XP0246E).

**Institutional Review Board Statement:** Not applicable.

**Informed Consent Statement:** Not applicable.

**Data Availability Statement:** Data are contained within the article.

**Acknowledgments:** The authors acknowledge the financial support of the Federal Ministry of Education and Research (BMBF) within the cluster project, "ProZell InnoRec". We acknowledge the financial support by the Open Access Publishing Fund of the Clausthal University of Technology.

**Conflicts of Interest:** The authors declare no conflicts of interest. No personal circumstances or interests that may be perceived as inappropriately influencing the representation or interpretation of the reported research results are given. The funders had no role in the design of the study; in the collection, analyses, or interpretation of data; in the writing of the manuscript; or in the decision to publish the results.

## Abbreviations

| | |
|---|---|
| LIBs | Lithium-ion batteries |
| EoL | End-of-life |
| NMC | Lithium nickel manganese cobalt oxides |
| HP-route | Hot pyrometallurgical route |
| WM-route | Warm mechanical route |
| CM-route | Cold-mechanical route |
| BM | Black mass |
| LE | Leaching efficiency |
| RT | Room temperature |
| IFAD | Institute of Mineral and Waste Processing, Recycling and Circular Economy Systems |
| IME | Institute of Process Metallurgy and Metal Recycling |
| iPAT | Institute for Particle Technology |
| MEET | MEET Battery Research Center |
| MVTAT | Institute of Mechanical Process Engineering and Mineral Processing |
| L/S | Liquid-to-solid |
| MIBC | Methyl isobutyl carbinol |
| EC | Ethylene carbonate |
| EMsC | Ethyl methyl carbonate |
| VC | Vinylene carbonate |
| GC-FID | Gas chromatography–flame ionization detection |
| IC-CD | Ion chromatography–conductivity detection |
| ACN | Acetonitrile |
| PC | Propylene carbonate |

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
