# Peer review of "The InnoRec Process: A Comparative Study of Three Mainstream Routes for Spent Lithium-ion Battery Recycling Based on the Same Feedstock"

_sustainability, doi:10.3390/su16093876_

Round 1

Reviewer 1 Report

Comments and Suggestions for Authors

This article aims to simplify existing recycling processes resulting from continuous changes in battery materials and battery design by simplifying existing purification technologies into three recycling pathways: pyrometallurgical hot path, mechanical hot path, and mechanical cold path. Additionally, this article compares the advantages and disadvantages of the three proposed ways. This article requires some corrections before it can be published.

1. In the abstract part of the study, some important numerical ratios of the three proposed models should be included. Emphasis should be placed on the contribution of this article's proposal to existing recycling processes.

2. The references in Table 1 do not need to be repeated in each cell. A reference tab can be placed for each row.

3. Sentences between lines 63-65 should be corrected.

4. Limits of the study should be included.

5. The conclusion part of the study should be enriched.

Comments on the Quality of English Language

Minor editing of English language required

Reviewer 2 Report

Comments and Suggestions for Authors

Recently, the importance of recycling has increased due to the increase in the cost of raw materials for lithium-ion batteries. This demonstrates the importance of the study. The article will be improved with the following corrections.

1. Between lines 63-65, there is a shift to the left.

2. Especially in articles where abbreviations are so dense, Nomenclature is required. I recommend you add it.

3. In Figure 6, the graphs should be aligned on the horizontal axis.

4. If the font size of the texts in Figures 14-16 is enlarged, they will be read better.

5. Innorec Project takes place in many places, but information about the details of the project should be given.

6. The Greenbat project is mentioned only in the conclusion and may cause information confusion for those who read it for the first time. A brief information should be given about this term.

Reviewer 3 Report

Comments and Suggestions for Authors

In this manuscript, the author presents a comprehensive comparative study of three primary recycling routes for waste lithium-ion batteries (LIBs), with the objective of identifying more efficient and resilient battery recycling methods. The study focuses on the recovery of six key elements: aluminum, copper, carbon, lithium, cobalt, and nickel. Additionally, the manuscript highlights the potential of the InnoRec process in enhancing recycling efficiency and fostering a sustainable circular economy for LIBs. Through the summarization of key research insights, the proposal of future research directions, and the optimization of LIB recovery processes, conclusive findings have been drawn. In my opinion, this paper can be accepted after minor revision.

1) The author should acknowledge that EoL NMC-622 LIB used is not a new battery. The author only compared the advantages and disadvantages of three different LIB recycling combination processes, and did not reflect the advantages of these three combination processes in new battery recycling. The author should re-condensate the innovative points of this manuscript (page 2 Line 63).

2) Please indicate the meaning of “InnoRec process”.

3) There are some errors in this paper. For example,3.4. Mechanical warm route” (page 9 Line 333) should be 3.4. Mechanical cold route” (page 9 Line 333).

4) The data of metal Mn yield  should be added in this manuscript. For example, Table 3.

5) The following articles might be helpful for understanding the research method if you cite in the manuscript.

Journal of Hazardous Materials, 461(2024): 132219.

ACS Sustainable Chemistry & Engineering, 9(5), 2271-2279.

Comments on the Quality of English Language

      6) The authors should thoroughly proofread the manuscript for grammatical errors.

Reviewer 4 Report

Comments and Suggestions for Authors

On the Introduction:

The introduction section seems to be lacking some clarity. Is the intention of the paper only to compare the effectiveness of each route involved in recycling? Or do they have suggestions to improve existing processes? The Introduction is not as convincing as the title of the paper. The article is considerably long because of its descriptive nature but the introduction is short for such a long article and fails to establish the premise of the paper. The presence of a flow chart in the introduction simply adds more woes to the Introduction. It is better for the article if the authors could move this figure to the background section. In order to improve this section kindly state what has been achieved in this work and how the article is structured as this is more of a report and not strictly a scientific work.

The background section looks like a collection of excerpts from various journals, magazines and textbooks without projecting a clear perspective. However, the rest of the information presented in this article could be of high importance to the battery recycling community and it seems suitable for publication after a major overhaul.

On the main results:

The authors have conducted a variety of analysis on the slurries of EOL Li ion batteries but there is no description of the techniques they have used to validate the results. For instance, there is Fig 4 and table 4 that discuss about the composition of the slags. The question to the authors is how did they obtain this data? There is also a figure on XRD measurements but there is no description of how the measurements were carried out and on what. All these things matter in scientific publishing and it looks like the authors have paid little to no attention to these details.

In section 3.1 there are a number of methods mentioned by acronyms. Before setting up a abbreviation or an acronym it has to be described. For example Li Ion batteries are abbreviated as LIB in this manuscript but why there is no such description in this section.

The figures used in this paper are very inconsistent. Some of the figures are fine whereas the others require a great deal of polishing. Figure 4 for the XRD with open top, right and left with not very legible legends is one good example. Figure 9 with an open top needs to be corrected. Figure 7 can be improved. Either plot them in origin or organize the figure in a uniform way.

The rest of the figures look alright but the use of a black and white tone in many of them gives a slightly odd appearance.

Comments on the Quality of English Language

The language has to be improved. This can be demonstrated with the very first few sentences in the Introduction:

"With the gradual popularity of electric vehicles worldwide, the production of lith- 31 ium-ion batteries (LIBs) is increasing at an incredible growth rate. How to effectively han- 32 dle end-of-life (EoL) LIBs is thus attracting more attention".

With the gradual increase in the popularity of electric vehicles worldwide, the production of lithium-ion batteries (LIBs) is increasing experiencing at an incredible growth . How to effectively handle. Therefore, the effective management and safety diposal of the end-of-life (EoL) LIBs becomes a crucial task and as a result this area is ganing attracting considerable attention than before.

Round 2

Reviewer 4 Report

Comments and Suggestions for Authors

All the drawbacks which were pointed out in the first round of revision have been answered sufficiently. The manuscript has been greatly improved. Very minor English editing could be required (but its optional) but speaking about  scientific competence, the manuscript is ready to be published in its present form barring some corrections during the publishing stage.